# Smoking-, Alcohol-, and Age-Related Alterations of Blood Monocyte Subsets and Circulating CD4/CD8 T Cells in Head and Neck Cancer

**DOI:** 10.3390/biology11050658

**Published:** 2022-04-25

**Authors:** Christian Idel, Kristin Loyal, Dirk Rades, Samer G. Hakim, Udo Schumacher, Karl-Ludwig Bruchhage, Ralph Pries

**Affiliations:** 1Department of Otorhinolaryngology, University of Luebeck, 23538 Luebeck, Germany; christian.idel@uksh.de (C.I.); kristin.loyal@uksh.de (K.L.); karl-ludwig.bruchhage@uksh.de (K.-L.B.); 2Department of Radiation Oncology, University of Luebeck, 23538 Luebeck, Germany; dirk.rades@uksh.de; 3Department of Maxillofacial Surgery, University of Luebeck, 23538 Luebeck, Germany; samer.hakim@uksh.de; 4Department of Anatomy and Experimental Morphology, University of Hamburg, 20251 Hamburg, Germany; u.schumacher@uke.de

**Keywords:** smoking, alcohol abuse, aging, head and neck cancer, monocytes, T cells, PD-L1, CX3CR1, CD11b

## Abstract

**Simple Summary:**

The formation of head and neck squamous cell carcinoma (HNSCC) is closely associated with cigarette smoking, alcohol abuse, and aging, but the immunologic alterations of peripheral monocytes and T-cells within this process are not fully understood. We aimed to investigate the impact of these factors on the abundances and characteristics of monocytes and CD4/CD8 T-cell subsets in the peripheral blood. Our data revealed distinct impacts of the individual behavior and age on the dynamics of these circulating immune cell subsets in head and neck cancer patients.

**Abstract:**

Head and neck squamous cell carcinoma (HNSCC) represents a heterogeneous malignant disease of the oral cavity, pharynx, and larynx. Although cigarette smoking, alcohol abuse, and aging are well-established associated factors for HNSCC, their respective influence on immunologic alterations of monocyte subsets or T-cell compositions in the peripheral blood has not yet been fully unveiled. Using flow cytometry, whole blood measurements of CD14/CD16 monocyte subsets and analyses of T-cell subsets in isolated PBMC fractions were carried out in 64 HNSCC patients in view of their tobacco and alcohol consumption, as well as their age, in comparison to healthy volunteers. Flow cytometric analysis revealed significantly increased expression of monocytic CD11b, as well as significantly decreased expression levels of CX3CR1 on classical and intermediate monocyte subsets in smoking-related and in alcohol-related HNSCC patients compared to healthy donors. Peripheral monocytes revealed an age-correlated significant decrease in PD-L1 within the entirety of the HNSCC cohort. Furthermore, we observed significantly decreased abundances of CD8^+^ effector memory T cells in active-smoking HNSCC patients and significantly increased percentages of CD8^+^ effector T cells in alcohol-abusing patients compared to the non-smoking/non-drinking patient cohort. Our data indicate an enhanced influence of smoking and alcohol abuse on the dynamics and characteristics of circulating monocyte subsets and CD4/CD8 T-cell subset proportions, as well as an age-related weakened immunosuppression in head and neck cancer patients.

## 1. Introduction

Head and neck squamous cell carcinoma (HNSCC) includes diverse groups of tumors of the oral cavity, pharynx, and larynx [1]. The causes of HNSCC are not fully understood; however, many different factors are closely associated with the occurrence of HNSCC, including tobacco, alcohol, diet, as well as infection with the human papillomavirus (HPV) [2,3,4]. Alterations of different immune cell subsets are associated with different clinical and biological features such as stage, site of disease, and especially age [5,6,7]. It has recently been shown that patients with more than 20 pack-years of smoking history exhibit a dose-dependent increased risk of development and recurrence of HNSCC and carry a distinct pattern of genomic alterations compared to non-smokers [8]. Generally, epidemiological studies suggest that about 60–80% of head and neck cancers are associated with cigarette smoking and/or alcohol abuse [9,10].

Tobacco smoking is known to influence the levels of immune cell populations such as CD8^+^ T lymphocytes, but its effect on specific immune cell subsets remains poorly understood [11]. Recent data suggest that active tobacco use in HNSCC has an immunosuppressive effect via an inhibition of tumor infiltration of cytotoxic T cells in comparison with never-smokers and former-smokers [12].

In addition to smoking, about 37% of the cancers of the oral cavity, the pharynx, and the larynx can be attributed to alcohol consumption [13]. Although the mechanisms for alcohol-related carcinogenesis are not fully understood, several studies revealed different carcinogenic effects such as impaired DNA repair mechanisms or functional protein alterations in response to ethanol and acetaldehyde [14,15,16].

Due to the risk accumulation over time, head and neck squamous cell carcinoma is a disease primarily affecting older individuals, with a median age at diagnosis of 66 years [17]. Age-related alterations of various immune cell subsets, termed ‘immunosenescence’, often goes along with increased levels of circulating inflammatory mediators such as C-reactive protein (CRP), interleukin (IL)-6, and tumor necrosis factor alpha (TNFα) [18,19]. Although it has been observed that there are no significant differences in the number of overall circulating monocytes in older adults [20], it has been proposed that the phenotypes of these cells are different. Monocytes are myeloid-derived cells that arise from hematopoietic stem cells in the bone marrow and belong to the mononuclear phagocyte system [21]. In general, monocytes can be subdivided in view of their CD14 and CD16 surface expression levels [22,23,24]. CD14^++^CD16^−^ ‘classical’ monocytes are classified as ‘naïve-like’ monocytes. The CD14^+^CD16^+^ ‘intermediate’ and CD14^dim+^CD16^+^ ‘non-classical’ monocyte subsets are considered to be more differentiated, activated, pro-inflammatory monocytes that exert specialized functions such as viral defense and patrolling behavior [25,26]. Under physiological conditions, they each comprise an amount of 5–10% of peripheral blood monocytes. An increase in these pro-inflammatory subsets has been shown in various acute and chronic inflammatory diseases [27,28]. Earlier studies identified that there was an increased frequency of CD16^+^ monocytes in older adults [29]. It has been proposed that the CD16^+^ monocytes are in fact a senescent monocyte population, with shorter telomeres, increased inflammatory potential, and expression of a senescence-associated microRNA [30,31]. All peripheral monocyte subsets are able to acquire macrophage morphology and characteristics, but exact differentiation potential of the different subsets remains incomplete [25].

However, although cigarette smoking, alcohol abuse, and age are well-established associated risk factors for head and neck cancer, their impact on cellular immunologic alterations in the peripheral blood of HNSCC patients has not yet been fully unveiled.

Thus, the aim of the present study was to investigate the impact of smoking, alcohol abuse, and age on differentiation patterns of circulating monocytes with regard to the three monocyte subsets described above. Therefore, first, a cohort of 64 HNSCC patients was subdivided into three distinct groups of non-smokers, former smokers, and active smokers, respectively. Furthermore, the cohort was subdivided in non-drinking and alcohol-abusing HNSCC patients as well as into two age-related groups of HNSCC patients ≤65 and ˃65 years of age, respectively. In addition, expression levels of monocytic fractalkine receptor CX3CR1 and integrin CD11b (Mac-1) were analyzed on circulating monocytes. The CX3CL1 receptor, CX3CR1, is involved in cell migration and adhesion and is required for monocyte homeostasis and survival [32,33]. CD11b is known to mediate binding to endothelial cells and migration through the vascular wall and has also been shown to act as a suppressor of immune responses in myeloid cells [34,35,36]. Furthermore, a detailed analysis of the CD4/CD8 T-cell subset composition was addressed to broaden our understanding on smoking-, alcohol-, and age-associated immunological alterations on a cellular level in the peripheral blood of HNSCC patients. 

## 2. Materials and Methods

### 2.1. Ethics Statement

All patients were treated surgically at the Department of Otorhinolaryngology, University Hospital Schleswig-Holstein, Campus Luebeck, and have given their written informed consent to participate in this study. The study was approved by the local ethics committee of the University of Luebeck (approval number 16-278) and conducted in accordance with the ethical principles for medical research formulated in the WMA Declaration of Helsinki.

### 2.2. Blood Collection and Patient Data

All blood donors provided written consent and were informed about the aims of the study and the use of their samples. Blood was drawn by venipuncture into a sodium citrate containing S-Monovette (Sarstedt; Nümbrecht, Germany). Blood samples were collected from healthy volunteers (*n* = 27; mean age of 59) and head and neck cancer patients (*n* = 64; mean age of 65). The clinicopathological characteristics of the patients are listed in Table 1.

### 2.3. Staining of Monocyte Subsets in Whole Blood

Within 4 h after blood collection, 20 µL of citrate blood was diluted in 80 µL PBS. Blood cells were stained with following antibodies: CD45-PE, CD14-FITC, CD16-BV-510, HLA-DR-APC-Cy7, CX3CR1-BV421, CD11b-BV421, and CD3-PerCP (all from Biolegend, San Diego, CA, USA). After 25 min staining in the dark, 650 µL RBC Lysis Buffer (Biolegend) were added to the samples and incubated for another 20 min. Subsequently, suspension was centrifuged at 400× *g* for 5 min and supernatant was discarded. The cell pellet was resuspended in 100 µL fresh PBS and used for FACS analysis.

### 2.4. Staining of T-Cell Subsets in Isolated PBMC

PBMC were isolated from the remaining blood by density gradient centrifugation in Biocoll (Biochrom GmbH, Berlin, Germany) at 400× *g* for 20 min. The upper PBS/plasma layer was removed and discarded. The PBMC layer was carefully harvested and transferred to a new 15 mL tube and washed once with PBS. The supernatant was discarded again and the PBMC pellet was resuspended in 1 mL PBS. For analysis of T-cell subsets, 100 µL of the cell suspension were incubated with the following antibody cocktail: CD3-PerCP, CD4-PE-Cy7, CD8-BV-510, PD-1-PE, PD-L1-APC, CD45RA-APC-Cy7, and CCR7-BV421. After 25 min staining in the dark, cells were washed with PBS and centrifuged at 400× *g* for 5 min. Cell pellets were resuspended in 200 µL PBS and used for FACS analysis.

### 2.5. FACS Analysis

Flow cytometry was performed with a MACSQuant 10 flow cytometer (Miltenyi Biotec, Bergisch-Gladbach, Germany) and data were analyzed using the FlowJo software version 10.0 (FlowJo, LLC., Ashland, OR, USA). All antibody titrations and compensations were performed in beforehand. For whole blood measurements, at least 100,000 CD45^+^ leukocytes were analyzed. Gating of monocyte subsets was performed as described before (Polasky et al., 2021). For T-cell analysis, 100,000 events within the PBMC gate were measured.

### 2.6. Statistical Analysis

Statistical analyses were performed with GraphPad Prism Version 7.0f (GraphPad Software, Inc., San Diego, CA, USA). The mean and standard error (SEM) are presented. The differences between groups were determined after testing for normal distribution and applying parametric (Student’s *t*-test), or non-parametric one-way ANOVA with Bonferroni post hoc test. The correlation between parameters was calculated using multivariate regression with the Pearson correlation coefficient. *p* < 0.05 (*), *p* < 0.01 (**), and *p* < 0.001 (***). Additional statistical details are given in the respective figure legends, when appropriate. Variations in replicate numbers for certain parameters were caused by blood samples which were too old for proper monocyte analysis on the one hand; or, on the other hand, by PBMC numbers which were too low for reputable T-cell subset characterization.

## 3. Results

### 3.1. Smoking-, Alcohol-, and Age-Related Monocyte Subset Distribution

The gating strategy of CD14/CD16-characterized monocyte subsets was carried out as described before [28]. In summary, CD45 was used as a pan leukocyte marker to facilitate whole blood measurement and monocytes were first roughly gated by their FSC/SSC characteristics and the positivity for CD14 and CD16. Neutrophil granulocytes, NK-cells, and B-cells were excluded by means of HLA-DR which is specific for monocytes. Remaining monocytes were then subgated into CD14^++^CD16^−^ (classical), CD14^++^CD16^+^ (intermediate), and CD14^dim+^CD16^+^ (non-classical) monocytes (Figure 1A). For flow cytometry analyses, HNSCC patients were subdivided into non-smoking, former-smoking (˃1 year), and active-smoking groups, respectively. In addition, the cohort was subdivided into non-drinking and alcohol-abusing patients and HNSCC patients ≤65 and ˃65 years of age. Our data revealed heterogeneous distributions of the abundances of the different monocyte subsets in HNSCC patients compared to healthy donors, but no significant differences between the defined smoking-, alcohol-, and age-related subgroups (Figure 1B–D).

In addition, PD-L1 expression from pan-monocytes was analyzed from isolated PBMC. Our data revealed significantly increased monocytic PD-L1 expression levels of overall HNSCC patients compared with healthy donors, but no significant differences between non-smoking, former-smoking, or active-smoking HNSCC patients; nor non-drinking and alcohol-abusing patients, respectively (Figure 2A). Surprisingly, HNSCC patients ˃ 65 years of age revealed significantly lower expression of monocytic PD-L1 compared to the younger patient cohort (*p* = 0.0379; Figure 2B). Correlation analysis of the entirety of the HNSCC patient cohort revealed an overall significant decrease in monocytic PD-L1 in correlation with increasing age (*p* = 0.0499 Figure 2C).

Furthermore, flow cytometric analysis revealed significantly increased expression of monocytic CD11b compared to healthy donors, with highly significantly increased levels in classical monocytes and intermediate monocytes from active-smoking patients (Figure 3A).

Similar results were obtained from the analysis of alcohol-abusing HNSCC patients compared to healthy donors (Figure 3C). Conversely, our data revealed significantly decreased expression levels of CX3CR1 on intermediate monocytes in active-smoking HNSCC patients compared to healthy donors (Figure 3B,D). In view of older HNSCC patients, data revealed an inverse regulation with significantly decreased expression of monocytic CD11b and increased expression of CX3CR1 in older HNSCC patients compared to the cohort of HNSCC patients ≤ 65 years of age (Figure 3E,F). A subsequent correlation analysis between monocytic CD11b and CX3CR1 expression of HNSCC patients revealed an overall highly significant negative correlation in all three monocyte subsets (Figure 4). 

### 3.2. Smoking-, Alcohol-, and Age-Related Distribution of T-Cell Subsets

We also analyzed the distribution of T cell subsets in head and neck cancer patients compared to healthy donors using flow cytometry. The T-cell differentiation from naïve to effector, effector memory, and central memory cells was analyzed for both CD4^+^ and CD8^+^ T cells by specific markers.

T-cell differentiation state was classified by means of CD45RA and CCR7 expression. We observed significantly decreased levels of CD4^+^ effector T cells accompanied by a significant increase in CD4^+^ effector memory T cells in HNSCC patients compared to healthy donors, whereas no significant differences were found between the smoking and alcohol-abusing subgroups (Figure 5).

No significant differences were found for percentages of naïve and central memory CD4^+^ T cells in HNSCC patients compared to healthy donors (Figure 5). 

Additional flow cytometric measurements revealed significantly decreased abundances of CD8^+^ effector memory T cells in active-smoking HNSCC patients compared to non-smoking patients, whereas no significant differences were found in the other analyzed CD8^+^ T-cell subsets in the smoking-related subgroups of HNSCC patients (Figure 6). Furthermore, significantly increased percentages of CD8^+^ effector T cells were identified in alcohol-abusing patients compared to the non-drinking subgroup (Figure 6).

Smoking and alcohol abuse are often occurring simultaneously in HNSCC patients, so we added an extra combined subgroup of smoking + alcohol-abusing patients within the analysis of CD8^+^ effector and effector memory T cells.

Our data revealed significantly increased percentages of CD8^+^ effector T cells in the smoking + alcohol-abusing subgroup compared to the non-smoking/non-drinking subgroup (Figure 6B). Significantly decreased abundances of CD8^+^ effector memory T cells were found in active-smoking HNSCC patients as well as in the smoking + alcohol-abusing subgroup compared to non-smoking/non-drinking patients (Figure 6B).

We observed significantly decreased levels of CD4^+^ effector T cells accompanied by a significant increase in CD4^+^ effector memory T cells in HNSCC patients compared to healthy donors, with no significant differences between the age-related subgroups. No significant differences were found in naïve and central memory CD4^+^ T cells (Figure 7A). Furthermore, flow cytometric analyses revealed slightly—but non-significantly—decreased percentages of naïve CD8^+^ T cells in HNSCC patients as well as no significant differences in the other analyzed CD8^+^ T-cell subsets, respectively (Figure 7B).

## 4. Discussion

### 4.1. Smoking-, Alcohol-, and Age-Related Alteration of Monocyte Subsets

Smoking, alcohol abuse, and aging are known to be closely associated with alterations of different immune cell subsets and the occurrence of head and neck cancer [5,37]. However, the impact of the individual lifestyle and life time of HNSCC patients on immune alterations of circulating monocytes and CD4/CD8 T-cell subsets is not fully unveiled yet. 

Our data revealed a noticeably heterogeneous monocyte subset distribution in HNSCC patients compared to healthy donors, but no significant smoking-, alcohol-, or age-related alterations were observed. Furthermore, we found increased levels of monocytic PD-L1 in HNSCC patients, but again no significant smoking- and alcohol-related differences. Our finding is in accordance with earlier observations in which increased expression levels of PD-L1 have been observed on CD14^+^CD16^+^ intermediate monocytes in HNSCC patients [38]. Surprisingly, however, we observed significantly decreased levels of monocytic PD-L1 in older HNSCC patients ˃ 65 years of age compared to the younger cohort.

Decreased PD-L1 expression in HNSCC patients ˃ 65 years of age could be due to an impaired activation of monocytic transcription factor NF-κB upon stimulation, as recently described in older adults [39].

PD-L1 is a checkpoint immunoregulatory molecule and involved in tumor escape mechanisms from T-cell immune responses [40]. In general, PD-L1 expression on myeloid cells has been correlated with poor prognosis of tumor patients [41,42,43]. Our findings corroborate earlier findings in patients with non-small cell lung cancer (NSCLC), that the expression of PD-L1 was not correlated with patients’ smoking histories [44].

Furthermore, we observed significantly increased expression of monocytic CD11b in all analyzed monocyte subsets compared to healthy donors with an increased significance in smoking-related HNSCC patients. It has been shown in mice that cigarette smoke exposure modified the composition of pulmonary macrophages by expanding CD11b^+^ subpopulations including monocyte-derived alveolar macrophages [45]. In addition, increased surface expression of CD11b was observed in smokers with chronic obstructive pulmonary disease (COPD) compared to non-smokers [46]. Taken together, these data suggest that cigarette smoke exposure leads to immune alterations of circulating myeloid cells which correspondingly results in altered compositions of tissue infiltrating macrophage subpopulations. Besides its function as a leukocyte adhesion molecule, integrin CD11b has also been shown to act as a suppressor of immune responses in myeloid cells [34,35,36].

Conversely, our data revealed significantly decreased expression levels of CX3CR1 on classical and intermediate monocyte subsets in smoking-related HNSCC patients, whereas no significant differences were found between former-smoking and active-smoking patients. Expression of CX3CR1 is involved in cell–cell interactions with the endothelium and thus required for monocyte crawling along the blood vessels [21]. Furthermore, CX3CR1 is known to be involved in atherosclerosis and vascular inflammation [47,48]. It has been shown to be expressed by all human monocyte subsets, although CD16^+^ monocytes express higher CX3CR1 levels. In view of older HNSCC patients, data revealed an inverse regulation with significantly decreased expression of monocytic CD11b and increased expression of CX3CR1 in all three monocyte subsets in older HNSCC patients compared to the cohort of HNSCC patients ≤ 65 years of age. However, little is known about the regulation of CX3CR1 at different stages during monocyte differentiation [33]. It is supposed that CX3CR1 plays a role in anti-tumor immune responses via the recruitment and activation of effector T lymphocytes [49,50]. In addition, increased levels of monocytic CX3CR1 expression in HNSCC patients ˃ 65 years of age may be involved in a known increased risk of cancer-associated thrombosis in older patients [51,52].

Though the versatile potential functions of integrin CD11b and fractalkine receptor CX3CR1 in view of smoking, drinking, and aging with regard to HNSCC progression remain elusive, its observed that altered expression on circulating monocytes in HNSCC patients most likely participates in the regulation of tumor immune suppression.

### 4.2. Alteration of Circulating T Cells

We also investigated the distribution of CD4^+^ and CD8^+^ T-cell subsets in the above mentioned subgroups of smoking- and alcohol-related head and neck cancer patients compared to healthy donors using flow cytometry. Tobacco smoking is known to impact circulating levels of major immune cells populations such as T cells, but its effect on specific immune cell subsets remains poorly understood [11]. Our investigations revealed significantly decreased percentages of CD4^+^ effector T cells accompanied with increased levels of CD4^+^ effector memory T cells in HNSCC patients, whereas no significant differences were found between the smoking- and alcohol-related subgroups. This observation underlines a tumor-driven immune suppression, since CD4^+^ T cells play a prominent role in tumor control through stimulation of different other immune cells such as CD8^+^ T cells or natural killer (NK) cells, respectively (75).

Remarkably, significantly increased levels of CD8^+^ effector T cells and decreased CD8^+^ effector memory T cells were found in active-smoking and alcohol-abusing HNSCC patients compared to non-smoking/non-drinking patients.

The importance of CD8^+^ T cells for anti-tumor immunity is well established and reflected by a series of prognostic analyses [53,54]. It has recently been shown that HNSCC patients who actively smoke have worse survival and have significantly lower numbers of CD8^+^ cytotoxic T-cells within the tumor immune microenvironment [12], which may explain the observed increased abundance of circulating CD8^+^ effector T cells in our study. In addition, smoking and alcohol abuse are supposed to increase the risk of HNSCC synergistically due to ethanol functioning as a solvent for smoking-related carcinogens into the mucosa of the upper aero digestive tract [55,56,57], which is reflected by the observed additive effect of smoking and alcohol abuse in circulating CD8^+^ T-cell abundances.

Furthermore, tobacco smoking during radiation therapy has been associated with unfavorable outcomes in head and neck cancer patients. It has been shown that smoking increases carboxyhemoglobin levels in the blood, which impairs hemoglobin ability to carry oxygen and thus results in poorer tumor oxygenation [58,59]. This negative effect of smoking during radiation therapy has been shown for different tumor entities such as bladder cancer, non-small-cell lung cancer, and HNSCC [58,60,61].

In addition, our investigations revealed significantly decreased percentages of CD4^+^ effector cells accompanied with increased levels of CD4^+^ effector memory cells, especially in older HNSCC patients. Memory T cells have the potential of long-term survival, while effector T cells are short-lived cells. In healthy individuals, memory T cells are quickly converted into large numbers of effector T cells upon activation [62]. Similar distributions from naive to effector memory T cells were observed in oropharyngeal cancer patients, independently of HPV status [63]. T-cell maturation occurs in the thymus and thus, age-related decrease in thymus function has been shown to result in a diminished T-cell generation and an accumulation of CD4^+^ memory T cells, respectively [64]. Furthermore, it has recently been shown that expression of immunosuppressive CD73 and CCR7 was lower on peripheral T cells in healthy volunteers as well as in tumor patients with increasing age [65].

## 5. Conclusions

Our data revealed a significant influence of the individual smoking and alcohol consumption behavior on the abundances and cellular characteristics of monocyte subsets and CD4/CD8 T cell subsets in the peripheral blood as well as an age-related weakened immunosuppression in head and neck cancer patients. Further comprehensive investigations on larger patient cohorts in correlation with the tumor immune infiltration and patient survival over a longer period of time are required to elucidate the relevance of the observed peripheral immune alterations for the clinical prognosis of patients suffering from this disease.

## Figures and Tables

**Figure 1 biology-11-00658-f001:**
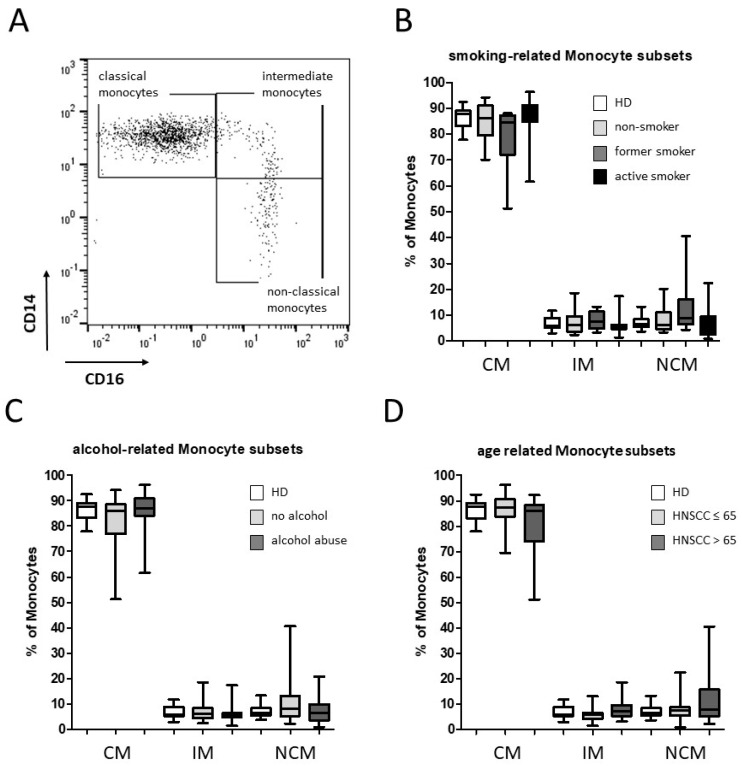
Flow cytometric analysis of CD14 and CD16 characterized monocyte subsets. (**A**) Representative example gating scheme of peripheral monocyte subset analysis by flow cytometry. (**B**) Whole blood analysis revealed similar median abundances of classical monocytes (CM), intermediate (IM), and non-classical monocytes (NCM) in non-smoking, former-smoking, and active-smoking HNSCC patients as well as in alcohol associated HNSCC patients (**C**) and age-related individuals (**D**) compared to healthy donors, but stronger dispersions of monocyte subset distributions in cancer patients.

**Figure 2 biology-11-00658-f002:**
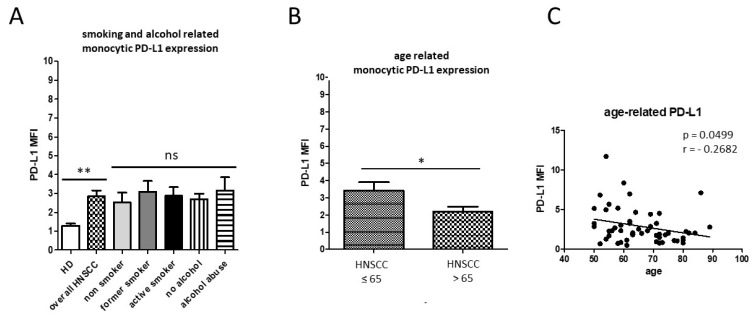
Flow cytometric analysis of PD-L1 expression. (**A**) HNSCC patients (*n* = 64) revealed significantly increased expression levels of monocytic PD-L1 compared to healthy donors, but no significant differences between the smoking- and alcohol-related subgroups (** *p* < 0.01; ns = not significant). (**B**) HNSCC patients ˃ 65 revealed significantly decreased expression levels of monocytic PD-L1 compared to HNSCC patients ≤ 65 (* *p* < 0.05). (**C**) Correlation analysis between monocytic PD-L1 expression and age of HNSCC patients revealed a significant negative correlation. The Pearson correlation coefficient (r) and *p* values are given. *p* < 0.05 was considered as significant.

**Figure 3 biology-11-00658-f003:**
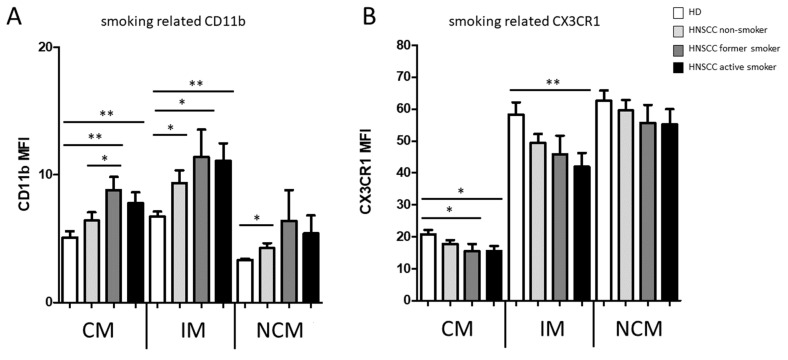
Flow cytometric analysis of monocytic CD11b and CX3CR1 expression in classical monocytes (CM), intermediate monocytes (IM), and non-classical monocytes (NCM) in PBMC from healthy donors and HNSCC patients. (**A**) Active-smoking HNSCC patients revealed significantly increased expression levels of monocytic CD11b compared to healthy donors. (**B**) HNSCC patients revealed significantly decreased expression levels of monocytic CX3CR1 in classical monocytes and intermediate monocytes from active-smoking patients compared to healthy donors. (**C**) HNSCC patients revealed significantly increased expression levels of monocytic CD11b compared to healthy donors. (**D**) HNSCC patients revealed significantly decreased expression levels of monocytic CX3CR1 in classical monocytes and intermediate monocytes compared to healthy donors. (**E**) HNSCC patients ≤ 65 revealed significantly increased expression levels of monocytic CD11b compared to healthy donors and HNSCC patients ˃ 65. (**F**) HNSCC patients ≤ 65 revealed significantly decreased expression levels of monocytic CX3CR1 compared to healthy donors and HNSCC patients ˃ 65. * *p* < 0.05; ** *p* < 0.01.

**Figure 4 biology-11-00658-f004:**
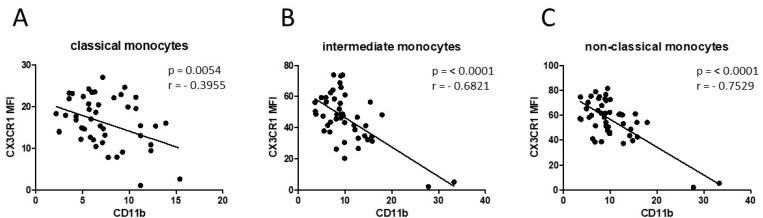
Correlation analysis between monocytic CD11b and CX3CR1 expression of HNSCC patients revealed a highly significant negative correlation in the analyzed (**A**) classical monocytes, (**B**) intermediate monocytes and (**C**) non-classical monocytes. A multivariate progression with the Pearson correlation was performed. The correlation coefficient (r) and *p*-values are given for each monocyte subset. *p* < 0.05 was considered as significant.

**Figure 5 biology-11-00658-f005:**
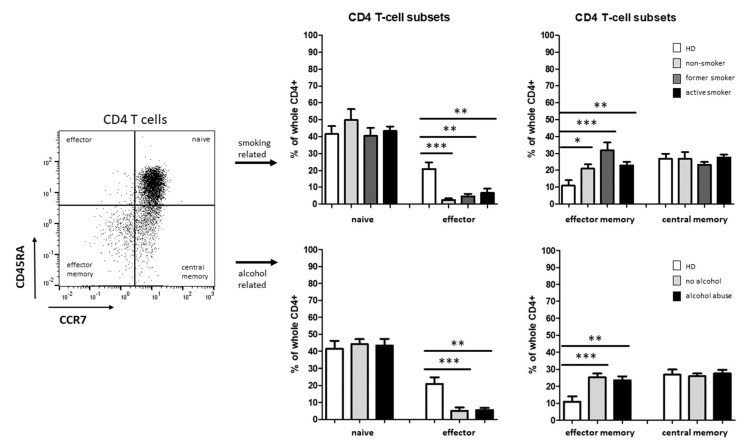
Flow cytometric analysis of CD4^+^ T-cell subsets in PBMC from healthy donors as well as smoking and alcohol-abusing HNSCC patients. Example gating scheme and percentages of naïve, effector, effector memory, and central memory T cells within CD4^+^ T cells. * *p* < 0.05; ** *p* < 0.01; *** *p* < 0.001.

**Figure 6 biology-11-00658-f006:**
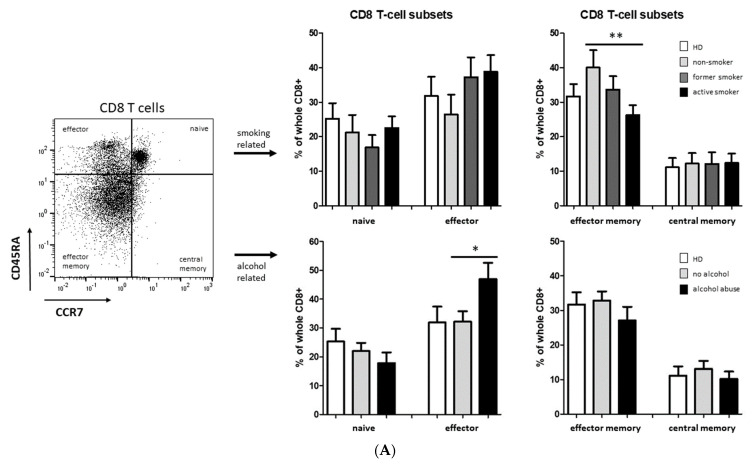
Flow cytometric analysis of CD8^+^ T cell subsets in PBMC from healthy donors as well as smoking- and alcohol-related HNSCC patients. (**A**) Example gating scheme and percentages of naïve, effector, effector memory, and central memory T cells within CD8^+^ T cells. (**B**) Flow cytometric analysis revealed significantly increased percentages of CD8^+^ effector T cells in smoking + alcohol abuse-related patients and significantly decreased percentages of CD8^+^ effector memory T cells in in active-smoking as well as in smoking + alcohol abuse-related patients compared to non-smoking/non-drinking patients. * *p* < 0.05; ** *p* < 0.01.

**Figure 7 biology-11-00658-f007:**
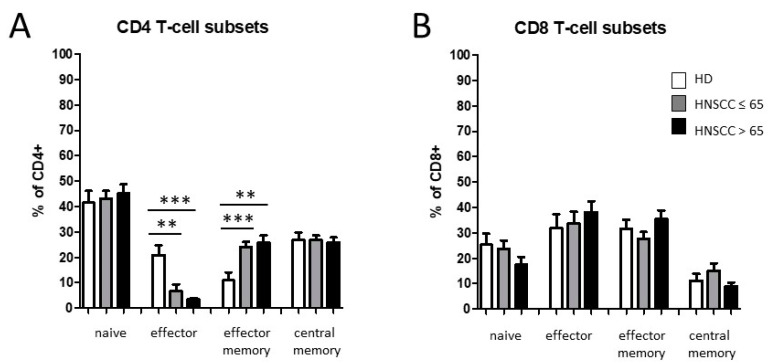
Flow cytometric analysis of CD4^+^ and CD8^+^ T cells in PBMC from healthy donors and HNSCC patients of two age-related cohorts (≤65 and ˃65 years of age, respectively). (**A**) Percentages of naïve, effector, effector memory, and central memory T cells within CD4^+^ T cells. (**B**) Abundances of naïve, effector, effector memory, and central memory T cells within CD8^+^ T cells. ** *p* < 0.01; *** *p* < 0.001.

**Table 1 biology-11-00658-t001:** Clinicopathological parameters.

Characteristics	Patients (*n* = 64)	
	*n*	%
**Gender**		
m	47	73
f	17	27
**Age (years)**		
≤65	38	59
>65	26	41
**Tumor Site**		
pharynx	37	58
larynx	13	20
oral cavity	14	22
**Tumor Stage**		
T1-T2	40	63
T3-T4	24	37
**HPV Status**		
positive	20	31
negative	44	69
**Alcohol Abuse**		
yes	14	22
no	50	78
**Tobacco Consumption**		
yes	45	70
no	19	30

## Data Availability

The data presented in this study are available on request from the corresponding author. The data are not publicly available due to the privacy policy of the University Hospital Schleswig Holstein.

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
