# Peer review of "Smoking-, Alcohol-, and Age-Related Alterations of Blood Monocyte Subsets and Circulating CD4/CD8 T Cells in Head and Neck Cancer"

_biology, 2022, doi:10.3390/biology11050658_

Round 1

Reviewer 1 Report

The paper, "Smoking, Alcohol, and Age-Related Alterations in Blood Mono-2 Cyte Subsets and Circulating CD4/CD8 T Cells in Head and Neck Cancer," is nicely written and includes all conceivable negative and positive controls. It is an excellent research paper.

Reviewer 2 Report

The manuscript has improved substantially after the modifications

This manuscript is a resubmission of an earlier submission. The following is a list of the peer review reports and author responses from that submission.

Round 1

Reviewer 1 Report

Smoking and Alcohol Abuse related Alterations of Blood Monocyte Subsets and circulating CD4/CD8 T Cells in Head and Neck Cancer

Comments:

The manuscript by Ralph Pries et al entitled " Smoking and Alcohol Abuse related Alterations of Blood Monocyte Subsets and circulating CD4/CD8 T Cells in Head and 3 Neck Cancer" describes the regulation of blood monocytes subsets and circulating CD4/CD8 T cells in the Head and Neck cancer. The authors start their analysis by testing the different monocyte subsets in HNSCC patients compared to healthy donors, but no significant differences between the defined smoking and alcohol related subgroups. Further they analyzed the expression of PD-L1 specific monocytes which show the significant upregulation in HNSCC in contrast to healthy individuals. This is not surprising, there is a mountain of evidence that suggest that the expression of PD-l1 is upregulated in HNSCC and as the result shows that it is not related to smoking and alcohol abuse. Expression of PD-L1 is not related to the stage-specific (http://ualcan.path.uab.edu/cgi bin/TCGAExResultNew2.pl?genenam=CD274&ctype=HNSC).

The manuscript did not show the molecular mechanism of differential expression of monocyte in HNSCC.

CD11b specific monocytes again did not show the direct connection with the smocking as active smokers did not show an increase in classical monocytes in contrast to the former smoker. In my opinion, this may be just because of the disease burden which shows the increase in the CD11b monocytes.

Overall I need to find any direct evidence of co-relation of smocking and differential expression of monocytes in HNSCC.

Reviewer 2 Report

The work describes the subsets of monocytes as well as, CD4/CD8 TL in smoking and/or alcohol abuse HNSCC patients. The manuscript is interesting, clearly written and very enjoyable to read. The figures are clear and facilitate the understanding of the results and analyzes carried out by the authors. However, some items should be modified to improve the manuscript:

  • Authors suggest a synergistic effect in the abstract. However, to affirm that, an adequate analysis is required (such as chou-talalai or bliss-independence method), since the effects could be additive and not synergistic. For this reason, it is more adequate employed enhancement/enhance and not synergism.

  • The authors indicate: “Conversely, our data revealed significantly decreased expression levels of CX3CR1 on classical and intermediate monocyte subsets in HNSCC patients compared to healthy donors, each with stronger effects in smoking and alcohol abusing HNSCC patients (Fig. 3B,D).” And: “Furthermore, we observed significantly increased expression of monocytic CD11b in all analyzed monocyte subsets compared to healthy donors with an increased significance in smoking related HNSCC patients”. Although there is a trend, there are no significant differences between non-smokers and smokers (formers and active) nor between non-drinkers and alcohol abusers. In this point, the only result with significant difference for CD11b is between non-smokers and former smokers in CM. For CX3CR1 the stronger effect is only a trend. Therefore, these assessments should be reformulated (throughout the manuscript) considering the statistical result and without assuming a trend as a result.

  • In figure 1C the bar corresponding to no alcohol NMC is darker than what is indicated in the legend (light gray).

Reviewer 3 Report

The authors suggested a synergic influence of smoking and alcohol abuse on the dynamic and characteristics of circulating monocyte subsets and CD4/CD8 T cell subsets in head and neck cancer patients. However, the logic of the current study was not complete, and the results were not sufficient to support the conclusions, so the manuscript was not of sufficient quality to be published in biology. Other issues for the authors in this paper are noted below.

  1. Traditional gating on CD14 and CD16 frequently led to contaminations of intermediate and non-classical monocytes; instead, the addition of markers, such as CD36, CCR2, HLA-DR, and CD11c enabled the more precise separation of human monocytes 
  2. There are many methodological problems and it is suggested that the authors review the results section statistically.
  3. What is the definition of a former smoker? Because the former smoker includes different times from one day to several years